# A Survey of Hepatitis B Virus and Hepatitis E Virus at the Human–Wildlife Interface in the Peruvian Amazon

**DOI:** 10.3390/microorganisms12091868

**Published:** 2024-09-10

**Authors:** María Fernanda Menajovsky, Johan Espunyes, Gabriela Ulloa, Stephanie Montero, Andres G. Lescano, Meddly L. Santolalla, Oscar Cabezón, Pedro Mayor

**Affiliations:** 1Departament de Sanitat i Anatomia Animals, Facultat de Veterinària, Universitat Autònoma de Barcelona, 08193 Bellaterra, Spain; mariafernanda.menajovsky@uab.cat; 2Unitat Mixta d’Investigació IRTA-UAB en Sanitat Animal, Centre de Recerca en Sanitat Animal (CReSA), Campus de la Universitat Autonoma de Barcelona (UAB), Catalonia, 08193 Bellaterra, Spain; johan.espunyes@irta.cat (J.E.); oscar.cabezon@uab.cat (O.C.); 3Institute of Agrifood Research and Technology (IRTA), Centre de Recerca en Sanitat Animal (CReSA), Campus de la Universitat Autonoma de Barcelona (UAB), Catalonia, 08193 Cerdanyola del Valles, Spain; 4Programa de Pós-Graduação em Saúde e Produção Animal na Amazônia, Universidade Federal Rural da Amazônia (UFRA), Belém 66077-830, Brazil; gulloau92@gmail.com; 5Emerge, Emerging Diseases and Climate Change Research Unit, School of Public Health and Administration, Universidad Peruana Cayetano Heredia, Lima 15015, Peru; stephanie.montero.t@upch.pe (S.M.); andres.lescano.g@upch.pe (A.G.L.); meddly.santolalla.r@upch.pe (M.L.S.); 6School of Medicine, Universidad Peruana de Ciencias Aplicadas (UPC), Lima 15067, Peru; 7Clima, Latin American Center of Excellence for Climate Change and Health, Universidad Peruana Cayetano Heredia, Lima 15024, Peru; 8Wildlife Conservation Medicine Research Group (WildCoM), Departament de Medicina i Cirurgia Animals, Universitat Autonoma de Barcelona, 08193 Bellaterra, Spain; 9ComFauna, Comunidad de Manejo de Fauna Silvestre en la Amazonía y en Latinoamérica, Iquitos 16006, Peru; 10Museo de Culturas Indígenas Amazónicas, Iquitos 16006, Peru

**Keywords:** epidemiology, hepatitis, HEV, HBV, risk factors, one health, tropical forests

## Abstract

Hepatitis B virus (HBV) and Hepatitis E virus (HEV) are zoonotic pathogens posing significant health concerns in rural Amazonia, a region marked by high endemicity, poverty, and limited healthcare access. However, the epidemiology of HBV and HEV in this ecosystem remains underexplored. This study examines the circulation of HBV and HEV at the human–wildlife interface and identifies risk factors within an isolated Amazonian indigenous community reliant on hunting for subsistence. Antibodies against HBV core antigens (HBcAbs) were found in three wildlife species: *Cuniculus paca* (0.8%), *Tayassu pecari* (1.6%), and *Mazama americana* (4.1%), marking the first record of HBV antibodies in free-ranging wildlife in the Amazon. However, further research is necessary to identify circulating strains and their relation to human HBV. HBcAbs were also detected in 9.1% of human samples, confirming exposure to HBV in the region. HEV IgG antibodies were present in 17.1% of humans and were associated with higher age. All wildlife and domestic animal samples tested negative for HEV, but transmission through consumption of wild animals and contaminated water needs further investigation. The identified risk factors highlight the urgent need for measures to promote safer food handling, improved sanitation, hygiene, and practices related to contact with wild animals.

## 1. Introduction

Hepatitis B virus (HBV) and Hepatitis E virus (HEV) are critical health concern worldwide, particularly in low- and medium-income countries (LMIC) [1,2,3]. In particular, in rural Amazonia, diverse areas of high endemicity for HBV and HEV have been identified, exacerbated by high levels of poverty and limited access to healthcare [4,5,6].

HBV, a DNA virus from the Family *Hepadnaviridae*, affects around one-third of the world population [7] and is responsible for approximately one million deaths annually due to cirrhosis, liver failure, hepatocellular carcinoma [8], and immune dysfunction linked to the intensification of other viral infections [9]. For decades, HBV has been hyperendemic in Amazonian rural populations, with a prevalence of over 50%, resulting in chronic active hepatitis and liver cirrhosis [4,10,11,12,13]. Transmission in these communities primarily occurs through direct contact with infected bodily fluids—including sexual intercourse and perinatal transmission—getting tattooed, and contact with non-indigenous populations [4,13,14]. This virus often exhibits genetic variants that are linked with specific hosts, but some studies indicate the presence of HBV and HBV-like in different wild mammals, suggesting the virus’s ability to infect and share hosts within their natural habitats [15,16,17,18]. While HBV transmission from wildlife reservoirs—often linked to bat bites, frequent in communities that regularly interact with wildlife—has been proposed and transmission between species of primates has been suggested [15,19], the transmission in rural communities with frequent contact with wildlife remains understudied [4]. 

HEV, an RNA virus from the Family *Hepeviridae*, is the main cause of acute viral hepatitis in humans worldwide and is associated with large outbreaks and epidemics in LMIC [20]. HEV is an emerging food-borne pathogen, generally transmitted through the handling and consumption of raw and undercooked infected meat and meat-derived products [21]. Acute outbreaks have also been reported in rural Amazonian communities due to lack of access to potable water and inadequate sanitation [3]. In South America, rural communities present seroprevalences ranging from 2.1% to 17% [22,23]. However, essential epidemiological aspects related to human zoonotic transmission still remain underexplored [6,23]. Domestic pigs and wild boars are considered the main reservoirs of HEV worldwide; however, the virus has also been identified in a wide range of wildlife species [24,25]. In Amazon rural communities, where animal farming and consumption of processed pork products are uncommon, there is a potential risk of HEV exposure from consuming infected wild meat or water and from direct contact with wild animals during hunting activities [6]. 

Despite the importance of HBV and HEV in the health of Amazonian rural communities, the functional role of wild mammals in HBV and HEV transmission through complex wildlife–human interactions facilitated by hunting and wild meat manipulation remains underexplored [26,27,28]. The present study aimed at evaluating HBV and HEV circulation in the human–wildlife interface and identifying risk factors and behaviors using a sociological analysis in an indigenous community that relies on subsistence hunting in a well-conserved and isolated area of the Peruvian Amazon. 

## 2. Materials and Methods

### 2.1. Study Area

This study was conducted in the Yagua indigenous community of Nueva Esperanza, located in the Yavarí-Mirín River basin (04°19′53″ S; 71°57′33″ W; UT5: 00), a geographically isolated and well-preserved forest along the border between Brazil and Peru in the Peruvian Amazon (Figure 1) [29]. This community of 370 people relies on a subsistence economy based on small-scale agriculture, hunting, and fishing [30,31]. The absence of feral pigs and livestock eliminates the possibility of disease transmission from these domestic sources [31]. Hepatitis-related mortality has been observed in other rural communities in the Yavarí River basin for over fifteen years, and since 2001, they have been grappling with a severe HBV outbreak [32]; 22 individuals died between 2001 and 2004, some with confirmed HBV and others with symptoms of hepatitis [33]. However, the lack of laboratory confirmation and serological data hinders authorities from addressing this situation [32]. 

Immunity against HBV in indigenous communities bordering Brazil and Peru in the Yavarí River basin is notably low due to lack of vaccination programs or significant delays between doses [32]. In the Peruvian Amazon, studies show that vaccination records are not well registered and serological tests do not align with reported vaccinations [34]. In the studied community, the indigenous population does not report any specific vaccination program against HBV, suggesting that the local population has not been appropriately vaccinated.

### 2.2. Blood Sample Collection

The blood sampling of wildlife took advantage of the discarded material from subsistence hunting, allowing for an extensive collection of 431 wild animals consumed by the local inhabitants between 2008 and 2020, including 125 peccaries (62 *Tayassu pecari* and 63 *Pecari tajacu*), 122 pacas (*Cuniculus paca*), 83 brocket deers (74 *Mazama americana* and nine *Mazama nemorivaga*), 66 primates (25 *Lagothrix poeppigii*, 15 *Sapajus macrocephalus*, 6 *Ateles chamek*, 6 *Pithecia monachus*, 5 *Cacajao calvus*, 5 *Cebus albifrons*, 2 *Saimiri macrodon*, 1 *Alouatta seniculus*, and 1 *Plecturocebus cupreus*), and 35 tapirs (*Tapirus terrestris*). In September 2019 and February 2020, a total of 43 peri-domestic rodents (38 rats and 5 mice) were also sampled. Blood samples were collected on either Whatman filter paper n. 3 or FTA^®^ cards (Scheilcher & Schuell, Dassel, Germany), preserved from 15 to 100 days in the community and later transported and stored at −70 °C, as previously reported [35,36]. 

Whole blood samples were also collected from 88 local residents, including 35 men and 53 women (39.8% vs. 60.2%, respectively), aged between 5 and 79 years, with a median age of 25 (10.0–34.0). Serum was extracted from the samples, stored in liquid nitrogen for transport, and then stored at −70 °C until laboratory analysis. 

### 2.3. Laboratorial Procedures

Blood-soaked filter papers containing the animal samples were processed by cutting a 132 mm^2^ piece, which was eluted in 400 µL of sterile phosphate-buffered saline (PBS) before vortexing for 20 s. The samples were stored at 4 °C for 24 h, then vortexed and frozen at −20 °C until analysis. The elutions from wild and peri-domestic animals and the human serum samples were tested for antibodies against HBV core antigens (HBcAbs) using the commercial ELISA kit “Human anti-hepatitis B virus core antibody” (Cusabio, Wuhan, China) [37]. HBcAbs are used to detect past or current HBV infections but does not indicate immunity from vaccination [38]. IgG antibodies against HEV were screened using the “ID Screen^®^ Hepatitis E Indirect Multi-species” kit (IDvet, Montpellier, France) for samples from wild and peri-domestic animals. For human blood samples, analysis for HEV antibodies was performed using the “Human hepatitis E virus antibody (IgG)” kit from Cusabio (Wuhan, China). Both commercial ELISA kits against HBcAb and HEVAb had intra- and inter-assay precision of CV% < 15%.

A recent study using the same samples on Whatman filter paper n. 3 and FTA^®^ cards revealed that only the DNA quantity and quality were adequate for molecular tests, whereas RNA was likely of limited use for viral pathogen research [39]. For this reason, only HBV DNA was extracted from the samples using the IndiMag Pathogen kit (Indical Bioscience, Leipzig, Germany), and the quality and quantity of DNA were determined using the Qubit dsDNA BR Assay Kit (Fisher Scientific, Waltham, MA, USA). A qPCR was conducted to detect HBV DNA using Promega PCR Master Mix, following the manufacturer’s instructions (Promega Corporation, Madison, WI, USA). The PCR analyzed the pre-S2/S region using the following primer pair and fluorescent probe purchased from Integrated DNA Technologies (IDT): forward primer (50-GAATCCTCACAATACCGCAGAGT-30), reverse primer (50-GCCAAGACACACGGGTGAT-30), and probe (50-FAM-AAGTCCACCACGAGTCTAG-NFQ/MGB-30) [25]. A plasmid carrying a human HBV 1.3 mergenome was used as a positive control. Given budget limitations, only 68 samples were analyzed, including all seropositive animals and humans.

### 2.4. Interviews for Risk Assessment 

In 2020, we used semi-structured surveys to collect data on habits and activities in the community to identify potential risk factors related to HBV and HEV infections. Interviews were conducted for 84 heads of families (47 (55.9%) women and 37 (44.1%) men, aged 18 to 77) from 42 different households (76.4% of total households). The questions were focused on activities usually related to the transmission of HBV, HEV, and other common bloodborne or foodborne pathogens, such as outdoor activities, especially hunting, but also contact with animals, the presence of domestic animals in households, and meat preparation and processing, as highlighted in previous studies [4,6,10,13,14,15,40,41]. The semi-structured surveys were divided into sections addressed to people based on their roles and activities within the community (See “Results section” for the detailed questions of the survey). Consequently, the number of respondents varied across different survey questions. By analyzing this information, we aimed to highlight practices that may expose individuals to HBV and HEV, allowing us to better understand the virus transmission and design appropriate prevention strategies [35]. 

### 2.5. Statistical Analysis 

A Generalized Linear Model (GLM) was employed to analyze the correlation between HBV and HEV antibody presence in wildlife and explanatory factors including species, habitat, and diet. The serological status (negative or positive) was utilized as the response variable, while “Habitat” (terrestrial/arboreal) and “Diet” (herbivorous/frugivorous/omnivorous/carnivorous) were considered as fixed explanatory variables.

Two GLMs were employed to examine the effect of age (in years) and sex (and their interaction) in relation to the serological results for HBV and HEV in humans. The response variable was the serological result, categorized as positive or negative. Model selection was based on the Akaike Information Criterion (AIC) [42]. 

GLM was also used to analyze the data collected in surveys on habits and activities in the community, with the aim of identifying potential risk factors related to HBV and HEV seropositivity. The serological status (negative or positive) was utilized as the response variable. 

All data analyses were performed using R 4.2.2 [43], and we considered a Type I error probability of 0.05 for hypothesis testing. 

## 3. Results

### 3.1. Serological Analysis

HBcAbs were only detected in three wildlife species: *Cuniculus paca* (0.8%; 1/122, 95% CI 0.1–0.5%), *Tayassu pecari* (1.6%; 1/62, 95% CI 0.3–8.6%), and *Mazama americana* (4.1%, 3/74; 95% CI 1.4–11.3%). The rest of the samples from wild animals were negative (Table 1). The serology of peri-domestic rodents was also negative, including rats (0/38; 0.0% CI 0.0–9.2%) and mice (0/5; 0.0% CI 0.0–43.5%). No significant association was observed between the HBV seroprevalence and biological or ecological factors (*p* > 0.05). 

In addition, HBcAbs were detected in 9.1% (8/88; 95% CI 4.7–16.9%) of the human samples. The models with the lowest AICs among all models were considered, specifically those with AIC differences of less than two, which included the models’ age (Akaike weight by age = 0.43), sex (Akaike weight by sex = 0.22), and ‘Sex + Age’ (Akaike weight by Sex + Age = 0.24) (Table 2); however, no significant association was observed between the HBV seroprevalence and age or sex (*p* > 0.05).

Antibodies against HEV were not found in wildlife (0.0%, 0/431, 95% CI 0.0–0.9%) or peri-domestic animals (0.0%, 0/43, 95% CI 0.0–8.2%), but were detected in 17.1% (15/88, 95% CI 10.6–26.4%) of humans. The models with the lowest AICs among all models were considered, specifically those with AIC differences of less than two, which included the models’ age (Akaike weight by age = 0.50) and Sex + Age (Akaike weight by Sex + Age = 0.33) (Table 3). The frequency of HEV seropositivity in humans increased with age (Estimate = 0.045, Std. Error = 0.018, z value= 2.264, *p* = 0.0104), but the influence of the sex of individuals was not statistically significant (*p* > 0.05). 

### 3.2. Molecular Analysis

The sixty-eight samples analyzed by conventional PCR to detect HBV DNA included sixty seronegative samples (twenty *Cuniculus paca*, twenty *Mazama americana*, ten *Pecari tajacu*, and ten *Tayassu pecari*) and eight seropositive human samples. All samples resulted negative. 

### 3.3. Risk Factors

The semi-structured surveys revealed that all inhabitants consume wild meat, which is cooked with water from unsafe sources (rivers, rain, streams) as they do not have access to potable water (Table 3). As drinking water, the local population mainly consumes rainwater and previously sedimented river water, but without purification treatment. A significant portion of the community consume meat with macroscopic lesions, meat prepared at low temperatures, offal, and even animals found dead in the forest. Additionally, domestic (cats, dogs, and chicken) and wild animals frequently come into contact with residents, and bat and mouse bites are common. In terms of safety practices, hunters admitted to not using protective measures when handling hunted animals, with some reporting injuries. Furthermore, a notable portion of the population does not use condoms for reproductive control and/or to prevent the spread of sexually transmitted infections. Table 4 summarizes other activities associated with HBV and HEV exposure.

## 4. Discussion

The present study improves the knowledge of the occurrence of HBV and HEV in humans and coexisting wildlife in the rural Peruvian Amazon. Antibodies against HBV were detected in three wild mammal species and both HEV and HBV antibodies were detected in humans. To our knowledge, this is the first record of antibodies against HBV in free-ranging wildlife in the Amazon region.

In the non-Amazon areas of Brazil, previous research has shown the occurrence of a virus phylogenetically close to human HBV sequences in domestic swine and wild boars (*Sus scrofa*), horses (*Equus ferus caballus*), domestic dogs (*Canis lupus familiaris*), jaguars (*Panthera onca*), maned wolves (*Chrysocyon brachyurus*), and crab-eating raccoons *(Procyon cancrivorus*) [18,26]. The presence of HBV antibodies across three distinct wildlife species in our study underscores the complexity of transmission, supporting the hypothesis that HBV or HBV-like viruses can infect different hosts in their natural habitats [15,26]. However, despite the high precision of the kits used, we cannot rule out the possibility of false negatives. On the other hand, further studies are required to determine which strains and genotypes are circulating in the Amazon and if they are closely related to human HBV, as previously observed in domestic pigs [18], and whether this virus is exchanged between animals and humans.

In the Amazon, human HBV prevalence ranges from 0 to 30% [44,45,46] and most studies found no clear gender-related nor age-related risk [18,47,48,49], which is consistent with our results. Our results confirm human exposure to HBV in the Yavarí-Marín River basin, stressing the need for the implementation of an appropriate vaccination program, highlighting the importance of extensive research across the Amazon to determine the prevalence and risk of HBV.

On the other hand, HEV circulation has been widely documented in domestic and wild fauna in South America [50,51,52]. In our study, all samples from wild and domestic animals were negative for antibodies against HEV. On the contrary, we observed a high seroprevalence of HEV in humans associated with age, as reported by previous studies [22,53], suggesting continuous and cumulative exposure to the virus over time [54]. However, RNA analysis, which could have provided more meaningful insights, was not performed due to the highly degraded RNA in our filter paper samples, as reported in a previous study using these same samples and demonstrating the greater degradation of RNA compared to DNA [39]. Our findings suggest that wild meat consumption is not a major factor in HEV transmission; however, this transmission through contact with or consumption of wild animals needs to be further explored, given the absence of interaction with pigs and consumption of pork products in the community. Our results also reinforce the idea that other environmental factors are causing repeated infection processes in humans. Some studies have demonstrated that HEV may be transmitted via the fecal–oral route from contaminated water and large waterborne outbreaks frequently occur, especially in developing countries [55,56,57]. However, the lack of reports on water-borne HEV outbreaks in the Amazon [50] evidences that this potential transmission route is not well studied. Further studies are needed to improve knowledge about the source of contamination, its occurrence, and its survival in water. 

Due to a low correspondence between serologically analyzed individuals and survey participants, the association between risk factors and serology results could not be analyzed. Despite this limitation, we have identified several common risk factors in Nueva Esperanza that may constitute a risk of HBV and HEV transmission according to the scientific literature [4,6,10,13,14,15,40,41]. This highlights the need to implement measures to promote safer food handling while also improving sanitation, hygiene, and practices related to close contact with wild animals and the purification of drinking waters. To address the hepatitis burden in the Yavarí River basin, it is critical to target specific risk factors. These results also underscore the need to explore and study cultural practices and animal reservoirs in the Amazon region, which may influence infectious disease pathways [58,59].

Traditionally, wildlife health data from isolated tropical environments has been restricted due to limited sampling capacity in such conditions; however, our long-term fieldwork in the study area, in collaboration with local people, allowed the collection of a large number of blood samples from wildlife. Previous studies demonstrated that serological tests are feasible with samples preserved on filter paper and that DNA could be preserved under these conditions [29,35,39], and we encourage researchers with similar samples to conduct serological and molecular DNA analyses to improve the understanding of virus circulation at the wildlife–human interface in remote Amazon environments. 

## 5. Conclusions

Our study provides evidence of past occurrences of HEV and HBV in humans in a remote Amazonian indigenous community. However, we were unable to confirm current infections due to a lack of molecular detection of both viruses. Additionally, we detected HBV antibodies in wild animals but could not clearly confirm the role of wildlife in the dissemination or harboring of these viruses, particularly in the foodborne transmission of HEV. To reach firmer conclusions, large-scale investigations with advanced molecular techniques are required. Nonetheless, the community’s traditional practices and behaviors, such as hunting and the eating wild meat, may significantly increase the risk of these diseases. Addressing these health risks through targeted interventions and public health initiatives is crucial to safeguarding the well-being of vulnerable rural communities that share such practices worldwide [40,41,60,61].

## Figures and Tables

**Figure 1 microorganisms-12-01868-f001:**
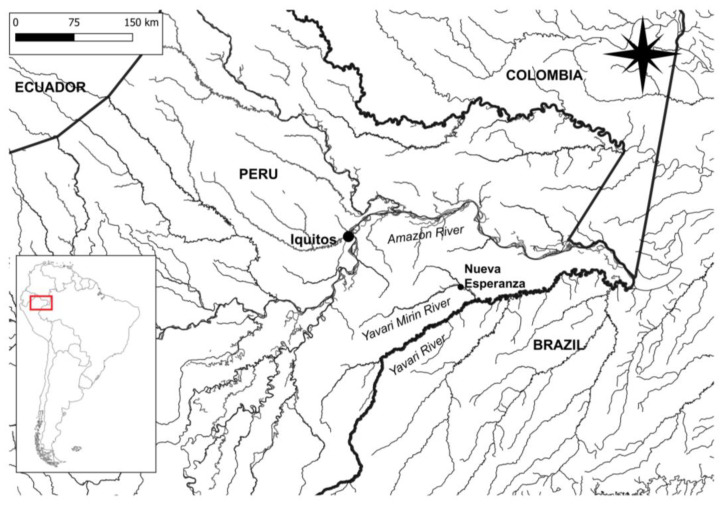
Location (red square) of the Nueva Esperanza community in the Yavari-Mirin River basin, a remote area on the border between Peru and Brazil, approximately 150 km far from Iquitos, the closest urban center. There is no accessibility to the study area through roads, only by river.

**Table 1 microorganisms-12-01868-t001:** Seroprevalence of HBcAbs among wild mammals hunted in Nueva Esperanza community (Peruvian Amazon) between 2008 and 2020.

Order, Family	Species	Tested	Positive (%)	95% CI
**O. Primates**		**66**	**0 (0.0%)**	**0.0–5.5%**
Atelidae	*Alouatta seniculus*	1	0 (0.0%)	0.0–79.4%
	*Ateles chamek*	6	0 (0.0%)	0.0–39.0%
	*Lagothrix l. poeppigii*	25	0 (0.0%)	0.0–13.3%
Pitheciidae	*Cacajao clavus*	5	0 (0.0%)	0.0–43.5%
	*Plecturocebus cupreus*	1	0 (0.0%)	0.0–79.4%
	*Pithecia monachus*	6	0 (0.0%)	0.0–39.0%
Callitrichidae	*Saimiri macrodon*	2	0 (0.0%)	0.0–65.8%
Cebidae	*Cebus albiforns*	5	0 (0.0%)	0.0–43.5%
	*Sapajus macrocephalus*	15	0 (0.0%)	0.0–20.4%
**O. Rodentia**		**122**	**1 (0.8%)**	**0.1–0.5%**
Cuniculidae	*Cuniculus paca*	122	1 (0.8%)	0.1–0.5%
**O. Cetartiodactyla**		**208**	**4 (1.92%)**	**0.75–4.8%**
Cervidae	*Mazama americana*	74	3 (4.1%)	1.4–11.3%
	*Mazama nemorivaga*	9	0 (0.0%)	0.0–29.9%
Tayassuidae	*Pecari tajacu*	63	0 (0.0%)	0.0–5.8%
	*Tayassu pecari*	62	1 (1.6%)	0.3–8.6%
**O. Perissodactyla**		**35**	**0 (0.0%)**	**0.0–9.9%**
Tapiridae	*Tapirus terrestris*	35	0 (0.0%)	0.0–9.9%
**Total**		**431**	**5 (1.16%)**	**0.5–2.7%**

**Table 2 microorganisms-12-01868-t002:** Candidate models considered in the study for HBV serology.

Candidate Models	k	AIC	Delta	w
Age	2	54.53	0.00	0.43
Sex	2	55.91	1.38	0.22
Sex + Age	3	55.74	1.21	0.24
Sex * Age	4	57.49	2.96	0.10

**Table 3 microorganisms-12-01868-t003:** Candidate models considered in the study for HEV serology.

Candidate Models	k	AIC	Delta	w
Age	2	77.17	0.00	0.50
Sex	2	81.48	4.31	0.06
Sex + Age	3	77.99	0.82	0.33
Sex * Age	4	80.08	2.91	0.12

**Table 4 microorganisms-12-01868-t004:** Answers to the semi-structured survey conducted to 84 heads of families from the Nueva Esperanza community in the Peruvian Amazon. The determination of risk factors is based on the literature: HBV [4,10,13,14,15] and HEV [6,40,41].

Activity	Frequency	HBV Risk [4,9,12,13,14]	HEV Risk [6,39,40]
Consumption of wild meat	100%, 84/84	No	Yes
Consumption of viscera from wild animals	55.89%, 19/34	No	Yes
Consumption of animals found dead in the forest	15%, 6/40	No	Yes
Use of safety measures when handling hunted animals	0%, 0/17	Yes	Yes
Inspection of lesions in hunted animals	41.17%, 7/17	No	Yes
Consumption of meat with lesions	82.5%, 33/40	No	Yes
Injuries while handling hunted animals	18.75%, 3/16	Yes	Yes
Preparation of meat at low temperatures	47.72%, 21/44	No	Yes
Source of drinking water supply		No	Yes
- Public water system	0%, 0/47		
- River	57.4%, 27/47		
- Rain	87.2%, 41/47		
Drinking water treatment		No	Yes
- No treatment (sedimentation)	76.1%, 35/46		
- Treatment	28.3%, 13/46		
Source of water used for cooking		No	Yes
- Rivers	82.4%, 28/34		
- Rainwater	52.9%, 18/34		
Wastewater disposal after cooking		No	Yes
- Household yard	26/34, 76.5%		
- River	9/34, 26.5%		
- Rustic drainage systems	3/34, 8.8%		
- Forest	1/34, 2.9%		
Use of animal products as medicine	19.35%, 12/62	Yes	Yes
Consumption of non-potable water	100%, 84/84	No	Yes
Presence of domestic animals at home	91.3%, 42/46	Yes	No
- Cats	19%, 8/42		
- Dogs	26.2%, 11/42		
- Chicken	100%, 42/42		
Presence of wild animals at home	25%, 19/76	Yes	Yes
- Birds (*Brotogeris versicolurus*)	73.7%, 14/19		
- Monkeys	5.3%, 1/19		
- Peccaries (*Pecari tajacu*)	15.8%, 3/19		
- Others	5.3%, 1/19		
Presence of mice and rats at home	100%, 84/84	Yes	Yes
Experienced bites from mice and rats	16.7%, 14/84	Yes	No
Presence of bats at home	98.8%, 83/84	Yes	No
Experienced bites from bats	26.2%, 22/84	Yes	No
Has tattoos	26.15%, 17/65	Yes	No
Has had surgeries or transfusions	27.41%, 17/62	Yes	No
Use of condoms	34%, 27/73	Yes	No

Due to the low individual correspondence between serological results and behavioral factors, no association was made between risk factors and HEV and HBV serology.

## Data Availability

The data presented in this study are available on request from the corresponding author. The data are not publicly available because it includes confidential information from people who participated in the study, even if they are anonymous.

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
