# Peer review of "A Survey of Hepatitis B Virus and Hepatitis E Virus at the Human–Wildlife Interface in the Peruvian Amazon"

_microorganisms, 2024, doi:10.3390/microorganisms12091868_

Round 1

Reviewer 1 Report

Comments and Suggestions for Authors

The manuscript "Survey of Hepatitis B Virus and Hepatitis E Virus at the human-wildlife interface in the Peruvian Amazon" shows interesting results that can contribute to the HBV and HEV epidemiology in the Amazon region. Nevertheless, there are some points to be considered.

1.Viral families should be written in italics.

2.Table 3 caption mentions HEV twice, please check.

3.Table 3: Use of safety measures when handling hunted animals, Injuries while handling hunted animals, Presence of wild animals at home and Presence of mice and rats at home should be considered an HEV risk.

4.It would be interesting to analyze by PCR the presence of a housekeeping gene in the collected samples in order to evaluate its viability, since no viral genome could be recovered in any case.

Author Response

Response to Reviewer 1 Comments

1. Summary

Thank you for your thorough review of our manuscript. We have carefully considered your feedback and have made the necessary revisions, which are clearly marked with track changes and highlighted in the re-submitted document.

2. Questions for General Evaluation

Reviewer’s Evaluation

Does the introduction provide sufficient background and include all relevant references?

Yes

Is the research design appropriate?

Can be improved

Are the methods adequately described?

Can be improved

Are the results clearly presented?

Can be improved

Are the conclusions supported by the results?

Can be improved

3. Point-by-point response to Comments and Suggestions for Authors

GENERAL COMMENT: The manuscript "Survey of Hepatitis B Virus and Hepatitis E Virus at the human-wildlife interface in the Peruvian Amazon" shows interesting results that can contribute to the HBV and HEV epidemiology in the Amazon region.

Comment 1: Viral families should be written in italics.

Response 1: Thank you for pointing this out. We have now edited this information in the manuscript.

Comment 2: Table 3 caption mentions HEV twice, please check.

Response 2: Thank you for pointing this out. We have now edited information on the table in the manuscript. Table 3 became Table 4.

Comment 3: Table 3: Use of safety measures when handling hunted animals, Injuries while handling hunted animals, Presence of wild animals at home and Presence of mice and rats at home should be considered an HEV risk.

Response 3: We agree with the reviewer. We have accordingly revised and modified this information in Table 4 (Table 3 became Table 4).

Comment 4: It would be interesting to analyze by PCR the presence of a housekeeping gene in the collected samples in order to evaluate its viability, since no viral genome could be recovered in any case.

Response 4: We based our choice of methodology on the results from a previous study that used the same samples as ours (Li et al., 2024). That study evaluated the effectiveness of identifying viral sequences and determined that the methods had limited applicability in viral pathogen research, especially for RNA viruses. Unfortunately, we could not perform any additional tests.

We agree and we have clarified this point at the Discussion section (page 8, paragraph 2, line 6): “However, RNA analysis, which could have provided more meaningful insights, was not performed due to the highly degraded RNA in our filter paper samples, as reported in a previous study using these same samples and demonstrating the greater degradation of RNA compared to DNA [39].”

Li, J., Ulloa, G. M., Mayor, P., Santolalla Robles, M. L., & Greenwood, A. D. (2024). Nucleic acid degradation after long‐term dried blood spot storage. Molecular Ecology Resources, e13979.

Reviewer 2 Report

Comments and Suggestions for Authors

Hepatitis B virus (HBV) and hepatitis E virus (HEV) are zoonotic pathogens-

The prevalence of this disease may pose serious health problems for rural areas in the Amazon region, so the investigation report in this article will better warn and prevent the spread of this disease in the area.However, there are still some issues in the article that need to be addressed.

1. Zhao ZY, Tang HC, Li F. Measles-associated severe pneumonia in a patient with HBeAg-negative chronic hepatitis B: A case report. Zoonoses, 2022, 2(1): 3. DOI: 10.15212/ZOONOSES-2021-0013, I suggest the author cite this literature.

2. The blood sampling of wildlife took advantage of the discarded material from sub-sistence hunting, allowing for an extensive collection of 431 wild animals consumed by the local inhabitants between 2008 and 2020, Why is this blood sample only available until 2020? This article has only been submitted now.

3. Human hepatitis E virus-How to avoid the high false positive rate of antibody (IgG) test kits?

4. A qPCR was conducted to detect HBV DNA using Promega PCR Master Mix, But I didn't see the qPCR results displayed in the article?

5. The discussion section of the article needs to be improved.

Author Response

Response to Reviewer 2 Comments

1. Summary

Thank you very much for taking the time to review this manuscript. We have addressed all your comments and suggestions, and the revisions are clearly indicated with track changes and highlighted in the updated file.

2. Questions for General Evaluation

Reviewer’s Evaluation

Does the introduction provide sufficient background and include all relevant references?

Yes

Is the research design appropriate?

Yes

Are the methods adequately described?

Yes

Are the results clearly presented?

Yes

Are the conclusions supported by the results?

Yes

3. Point-by-point response to Comments and Suggestions for Authors

GENERAL COMMENT: The prevalence of this disease may pose serious health problems for rural areas in the Amazon region, so the investigation report in this article will better warn and prevent the spread of this disease in the area.

Comment 1: Zhao ZY, Tang HC, Li F. Measles-associated severe pneumonia in a patient with HBeAg-negative chronic hepatitis B: A case report. Zoonoses, 2022, 2(1): 3. DOI: 10.15212/ZOONOSES-2021-0013, I suggest the author cite this literature.

Response 1: Agree. We have added the reference (Page 2, paragraph 1, line 3): “HBV, a DNA virus from the Family Hepadnaviridae, affects around one-third of the world population [7] and is responsible for approximately one million deaths annually due to cirrhosis, liver failure, and hepatocellular carcinoma [8] and immune dysfunction linked to the intensification of other viral infections [9].”

Comment 2: The blood sampling of wildlife took advantage of the discarded material from subsistence hunting, allowing for an extensive collection of 431 wild animals consumed by the local inhabitants between 2008 and 2020, Why is this blood sample only available until 2020? This article has only been submitted now.

Response 2: Unfortunately, due to the remoteness of the study location and financial constraints, we worked in the study area for 12 years (2008-2020). In 2020, with the pandemic, the project was definitively interrupted. It took us a long time to obtain permission to export human and wildlife samples, and we had parallel activities. Sometimes we would like to be much faster than we can be.

Comment 3: Human hepatitis E virus-How to avoid the high false positive rate of antibody (IgG) test kits?

Response 3: Unfortunately, we were unable to use additional methods to double-check the results. However, we used a kit that includes appropriate negative and positive controls to identify non-specific signals (https://www.cusabio.com/ELISA-Kit/Human-Hepatitis-E-virus-antibodyIgGELISA-Kit-114601.html). This kit demonstrates high sensitivity and excellent specificity for detecting human hepatitis E virus antibody (IgG), with no significant cross-reactivity or interference from analogues reported (https://www.cusabio.com/uploadfile/newwell/Instructions/CSB-E04811h_Human_Hepatitis_E_virus_antibody(IgG)ELISA_Kit.pdf). In addition “Both commercial ELISA kits against HBcAb and HEVAb had intra- and inter-assay precision of CV% <15%.”,  (First paragraph in the 2.3. Laboratorial procedures) and also “However, despite the high precision of the kits used, we cannot rule out the possibility of false negatives.” (Second paragraph in the Discussion section)

Comment 4: A qPCR was conducted to detect HBV DNA using Promega PCR Master Mix, But I didn't see the qPCR results displayed in the article?

Response 4: We mentioned the negative results of this test in Result Section 3.2. Molecular analysis: “The 68 samples analyzed by conventional PCR to detect HBV DNA included 60 ser-onegative samples (20 Cuniculus paca, 20 Mazama americana, 10 Pecari tajacu, and 10 Tayassu pecari) and eight seropositive human samples. All samples resulted negative.” (page 6, paragraph 2, line 1-3)

Comment 5: The discussion section of the article needs to be improved.

Response 5: Agree. We have accordingly revised the Discussion section. 

Reviewer 3 Report

Comments and Suggestions for Authors

In this study, authors investigated the epidemiology of HBV and HEV at the human-wildlife interface and identified the risk factors of transmission. The topic of this study is interesting and meaningful, and the manuscript is well-written. There are some minor suggestions.

1. It would be more meaningful if HEV RNA can be tested. Authors are suggested to mention this limitation in the discussion and discuss about it.

2. It is suggested that the information of animals and their results can be summarized as a table to make it clear.

3. Describe more about the point of “HBV as a zoonotic pathogen” in the introduction.

Author Response

Response to Reviewer 3 Comments

1. Summary

Thank you for providing such insightful feedback on our manuscript. We have incorporated your suggestions and have highlighted these changes and used track changes in the re-submitted version.

2. Questions for General Evaluation

Reviewer’s Evaluation

Does the introduction provide sufficient background and include all relevant references?

Can be improved

Is the research design appropriate?

Can be improved

Are the methods adequately described?

Yes

Are the results clearly presented?

Yes

Are the conclusions supported by the results?

Yes

3. Point-by-point response to Comments and Suggestions for Authors

GENERAL COMMENT: In this study, authors investigated the epidemiology of HBV and HEV at the human-wildlife interface and identified the risk factors of transmission. The topic of this study is interesting and meaningful, and the manuscript is well-written.

Comment 1: It would be more meaningful if HEV RNA can be tested. Authors are suggested to mention this limitation in the discussion and discuss about it.

Response 1: We agree and we have clarified this point at the Discussion section (page 8, paragraph 2, line 6): “However, RNA analysis, which could have provided more meaningful insights, was not performed due to the highly degraded RNA in our filter paper samples, as reported in a previous study using these same samples and demonstrating the greater degradation of RNA compared to DNA [39].”

Comment 2: It is suggested that the information of animals and their results can be summarized as a table to make it clear.

Response 2: Agree. We have included a table (Table 1) including these results.

Comment 3: Describe more about the point of “HBV as a zoonotic pathogen” in the introduction.

Response 3: The zoonotic nature of HBV has been suggested, and part of our study involves analyzing samples from both humans and animals simultaneously. Although we did not analyze genetic profiles, this approach can still provide valuable indicators of presence of the virus in the same environment. We have improved the Introduction section to provide account of the specific evidence supporting the zoonotic nature of the virus. (Page 2, paragraph 1, line 12): “While HBV transmission from wildlife reservoirs—often linked to bat bites, common in communities that regularly interact with wildlife— has been proposed, and transmission between species of primates has been suggested [15, 19], the transmission in rural communities with frequent contact with wildlife remains understudied [4]. “

Round 2

Reviewer 1 Report

Comments and Suggestions for Authors

All the comments were addressed by the authors